## COMMENT

# MacroGreen, a simple tool for detection of ADP-ribosylated proteins

Antonio Ginés García-Saura[1], Laura K. Herzog[2,4], Nico P. Dantuma[2] & Herwig Schüler [1,3✉]

Enzymes in the PARP family partake in the regulation of vital cellular signaling pathways by ADP-ribosylating their targets. The roles of these signaling pathways in disease development and the de-regulation of several PARP enzymes in cancer cells have motivated the pursuit of PARP inhibitors for therapeutic applications. In this rapidly expanding research area, availability of simple research tools will help assess the functions of ADP-ribosylation in a wider range of contexts. Here, we generated a mutant Af1521 macrodomain fused to green fluorescent protein (GFP) to generate a high-affinity ADP-ribosyl binding reagent. The resulting tool – which we call MacroGreen – is easily produced by expression in *Escherichia coli*, and can detect both mono-and poly-ADP-ribosylation of diverse proteins in vitro. Staining with MacroGreen allows detection of ADP-ribosylation at sites of DNA damage by fluorescence microscopy. MacroGreen can also be used to quantify modification of target proteins in overlay assays, and to screen for PARP inhibitors in high-throughput format with excellent assay statistics. We expect that this broadly applicable tool will facilitate ADP-ribosylation related discoveries, including by laboratories that do not specialize in this field.

Intracellular protein ADP-ribosylation catalyzed by the PARP enzymes is part of medically important signaling events.[1,2] The oncology drug target PARP1 activates DNA repair by giving rise to poly-ADP-ribose (PAR) chains on its target proteins and itself. PARP inhibitors are used to treat BRCA deficient cancers; but likely this class of compounds is underexplored. Both other PARylating family members such as the tankyrases (PARP5a, PARP5b) and mono-ADP-ribosylating (MARylating) family members, including PARP10 and PARP14, regulate cancer related processes and are de-regulated in certain cancers.[3–5] The ADP-ribosyl glycohydrolases that remove the modification are likewise of great medical interest.[6] The pharmacological inhibition of those enzymes is under scrutiny for new treatment opportunities.

Due to the significance of ADP-ribosylation, its detection and quantification has become an objective of great interest in biomedical research. Before the effects of PARP inhibitors can be fully appreciated, the identification and characterization of the ADP-ribosylated proteome is a priority. ADP-ribosylation can be detected after incorporation of radioisotopes or fluorescently

[1] Department of Biosciences and Nutrition, Karolinska Institutet, Huddinge, Sweden. [2] Department of Cell and Molecular Biology, Karolinska Institutet, Stockholm, Sweden. [3] Center for Molecular Protein Science, Department of Chemistry, Lund University, Lund, Sweden. [4] Present address: Department of Chemistry, Umeå University, Umeå, Sweden. ✉email: herwig.schuler@biochemistry.lu.se

labeled ADP-ribose; but these methods are not available in all laboratories and are limited in their applicability to certain PARP enzymes and applications. Instead, an antibody that binds to PAR oligomers consisting of at least 10 ADP-ribose units[7] has been the most widely used tool over decades. Recently, an antibody that can recognize both MAR and PAR has been presented[8] and antibodies that specifically recognize MARylated targets have been produced.[9]

Meanwhile, in our cells, target-linked ADP-ribosyl is recognized by several binding modules – protein domains involved in the cellular ADP-ribosylation signaling pathways[10] – and some of these have already been employed to detect ADP-ribosylation. One of a few proteins containing a macrodomain that binds to both PARylated and MARylated proteins[11] is the *Archaeoglobus fulgidus* macrodomain protein Af1521. Its affinity for free ADP-ribose is in the high nanomolar range ($K_D = 126$ nM)[12], which is together with MacroD2 the highest affinity known to date for any macrodomain.[13,14] DiGirolamo and co-workers first used Af1521 to enrich ADP-ribosylated proteins for identification by mass spectrometry[15] and later studies have used the same binder domain,[16–20] which has provided important insights into the cellular ADP-ribosylome. Kraus and co-workers[21] refined the concept by fusing the Fc region of rabbit immunoglobulin to the Af1521 macrodomain, to macroH2A, to a construct containing the three macrodomains of PARP14, as well as to a WWE domain. The study confirmed that Af1521 is a useful tool for the recognition of pan-ADP-ribosylation.[21] The Fc-fusion modules were applicable to immunoblotting, immunoprecipitation and immunofluorescent staining of cells. However, the method still required a secondary antibody and a mode of detecting it. This was overcome by Rabouille and co-workers, who detected PARylation and/or MARylation in vitro and in vivo using either human macroH2A or the three macrodomains of PARP14 fused to YFP.[22] However, the affinity of PARP14 macrodomains for MARylated proteins is low.[23]

Nowak and co-workers used in vitro evolution to optimize ADP-ribose binding in Af1521.[24] Although the affinity of evolved Af1521 (eAf1521) for modified target protein was not assessed, eAf1521 bound free ADP-ribose with an affinity of 3 nM, which is unprecedented among known natural binder domains. Comparative characterization, including target enrichment and identification by mass spectrometry, indicated that eAf1521 binds also ADP-ribosylated proteins with higher affinity than wild type Af1521. However, Af1521 also possess ADP-ribosyl glycohydrolase activity toward automodified PARP10.[13,25] The eAf1521 protein retained that activity,[24] imposing some method limitations. Here, to develop this research tool further, we fused the wild type Af1521 protein to GFP, introduced the mutations that were decisive for the superior affinity for free ADP-ribose in

eAf1521, and then introduced further mutations to reduce the ADP-ribosyl glycohydrolase activity of the protein. The resulting protein reagent, which we call MacroGreen, is easy to produce and suitable for rapid detection of ADP-ribosylated proteins in vitro without a need for specialist reagents and time-consuming methods.

## Results and discussion

**Development of the MacroGreen recombinant protein.** The aim of our study was to obtain a fluorescent Af1521 protein derivative that combined high-affinity ADP-ribosyl binding with low ADP-ribosyl glycohydrolase activity. To that end, we designed Af1521 mutants, which were expressed as GFP fusion proteins in bacteria. To assess the ADP-ribosyl binding capacity of Af1521 mutants, we employed a simple method in which MAR- and PARylated proteins were attached to multiwell plates, after which the macrodomain-GFP fusion proteins were overlain, and GFP fluorescence was quantified (Fig. 1). The crystal structure and biochemical analyses of the recently described eAf1521 indicated that the two amino acid substitutions K35E,Y145R were sufficient for nanomolar affinity to free ADP-ribose.[24] Indeed, the Af1521 protein with a C-terminal GFP tag and the eAf1521 derived K35E,Y145R replacement was similar to the eAf1521-GFP fusion construct in terms of both ADP-ribosyl binding and ADP-ribosyl glycohydrolase activity (Af1521-c002; Table 1 and Supplementary Fig. 1). This also showed that the GFP fusion did not affect the binding and enzymatic properties of the macrodomain; but it dramatically increased the yield of purified protein after overexpression in *E.coli*.

Next, we introduced a series of amino acid replacements, based on previous studies of homologous proteins, with the goal to reduce ADP-ribosyl glycohydrolase activity (Supplementary Fig. 2). These mutations were introduced in Af1521-c002, the construct with the eAf1521 derived K35E,Y145R replacement followed by a C-terminal GFP tag. The G123E mutation of human TARG abolished O-Ac-ADP-ribose hydrolase activity.[26] The corresponding mutation, G143E, reduced ADP-ribosyl glycohydrolase activity (Af1521-c003; Table 1); however, its binding to ADP-ribosylated PARP10 was also seriously impaired (Supplementary Fig. 1a). The N34A mutation of Af1521 resulted in a 3-fold decrease in affinity for free ADP-ribose.[12] Af1521-c004, containing the mutation N34S in addition to K35E,Y145R, did not differ significantly in glycohydrolase activity or target binding from Af1521-c002 (Table 1, Supplementary Fig. 1a). Af1521-c005, containing R36N, was designed to probe possible involvement of Arg36 in the catalytic mechanism, as its side chain aligns with the N34 side chain and coordinates a water molecule near the terminal ribose.[12,24] However, Af1521-c005 had properties similar

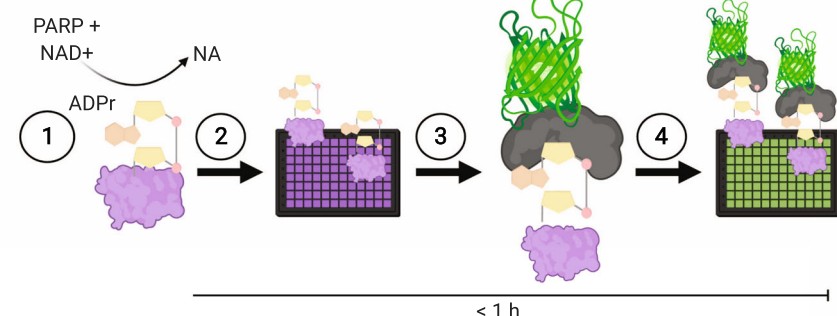

**Fig. 1 MacroGreen as a tool to detect MAR- and PARylation.** Overview of MARylated and PARylated protein overlay assays in multiwell plates and detection using MacroGreen fluorescence. The assay consists of (1) the ADP-ribosylation reaction; (2) binding of ADP-ribosylated protein to multiwell plates; (3) binding of MacroGreen to ADP-ribosylation sites; and (4) fluorescence readout using a plate reader. Graphics generated using BioRender.

| Construct | Mutations* | $k_{cat}$ ** | $R^2$ |
|---|---|---|---|
| **Table 1 Summary of Af1521-GFP fusion proteins evaluated in this study and their ADP-ribosyl hydrolase rate constants.** | | | |
| eAf1521 | **K35E**, Y74C, F97L, S105G, E110G, **Y145R**, N162D | 0.095 ± 0.007 | 0.9882 |
| Af1521-c001 | wild type | 0.153 ± 0.018 | 0.9638 |
| Af1521-c002 | **K35E**,**Y145R** | 0.107 ± 0.008 | 0.9935 |
| Af1521-c003 | **K35E**,G143E,**Y145R** | 0.051 ± 0.012 | 0.9483 |
| Af1521-c004 | N34S,**K35E**,**Y145R** | 0.098 ± 0.008 | 0.9904 |
| Af1521-c005 | **K35E**,R36N,**Y145R** | 0.097 ± 0.008 | 0.9907 |
| Af1521-c006 | G42E, I144R, Y145N | 0.146 ± 0.017 | 0.9778 |
| Af1521-c007 ("MacroGreen") | **K35E**, G42E, I144R, **Y145R** | 0.075 ± 0.007 | 0.9812 |
| Af1521-c008 | **K35E**, G42E, I144R, Y145N | 0.139 ± 0.018 | 0.9744 |

*Bold: eAf1521 mutations essential for nanomolar binding of free ADP-ribose
** Determined by fitting to a first order equation. $N = 2$ to 4; $n = 3$ (original data exemplified graphically in Supplementary Fig. 1b).

to Af1521-c002. Then, we designed a set of mutant constructs based on the Chikungunya virus (CHIKV) nsP3 macrodomain. The CHIKV nsP3 G32E,V113R,Y114N mutant retains 15% glycohydrolase activity and has a 3.5-fold higher affinity for ADP-ribose compared to the wild type protein.[27] The G42E mutation of Af1521 (homologous to CHIKV nsP3 G32E) has been shown to abolish ADP-ribose binding.[15] However, the crystal structure of eAf1521 showed that the K35E,Y145R replacement induced a shift in the terminal ribose[24] such that a glutamate in position 42 might be accommodated. Thus, we tested the homologous mutations G42E,I144R,Y145N (Af1521-c006). This protein had properties similar to the wild-type domain. However, when we combined this set of mutations with the eAf1521 derived mutations to obtain Af1521-c007, this protein showed an ADP-ribosyl glycohydrolase activity reduced to half of the wild type domain, and displayed slightly but significantly better binding to ADP-ribosylated targets compared to Af1521-c002 and eAf1521 (Table 1, Supplementary Fig. 1a,b). Finally, Af1521-c008, containing the Y145N mutation, had properties similar to the wild-type domain. We conclude that Af1521-c007, a GFP-fusion quadruple mutant containing the replacements K35E,G42E,I144R,Y145R, was a considerable improvement over the GFP-tagged wild-type domain as an ADP-ribosyl detection reagent, and we called this protein MacroGreen.

**MacroGreen detects an array of ADP-ribosylated substrates**. To further characterize the ADP-ribosyl binding properties of Macro-Green, we conducted overlay assays with different ADP-ribosylated proteins over a range of concentrations. As expected, based on the binding properties of wild type Af1521 protein, MacroGreen bound both PARylated PARP1 (Fig. 2a) and MARylated PARP10 (Fig. 2b). Comparison of MacroGreen with the wild type Af1521 protein fused to GFP showed that MacroGreen was eight times more efficient in detecting PARylated PARP1, and three times more efficient in detecting MARylated PARP10 (comparing linear slopes over the concentration ranges shown in Fig. 2a and b, respectively). We quantified the binding affinity of the interaction between Macro-Green and the auto-MARylated PARP10 catalytic domain by surface plasma resonance (SPR) measurements. MacroGreen protein was covalently immobilized and its affinity for MARylated PARP10 was measured. Based on the kinetics of the binding and unbinding events, each MacroGreen molecule bound to one molecule of MARylated PARP10 with an apparent $K_D$ value of 30 nM (Supplementary Fig. 3).

To test whether MacroGreen discriminated between ADP-ribosyl moieties linked to different acceptor residues, we conducted concentration response assays with four different substrates. PARP1 alone auto-PARylates predominantly on carboxylic acid side chains;[28] PARP1 in presence of HPF1 auto-MAR- and -PARylates predominantly on serine side chains;[29]

PARP10 auto-MARylates on glutamic and aspartic acid, serine, lysine and arginine side chains;[27,30–33] and actin is MARylated at a single arginine residue by the *Clostridium botulinum* toxin C2I subunit.[34] In all four cases, the MacroGreen assay showed a linear response over a wide range of substrate concentrations (Fig. 2c). For auto-MARylated PARP10, the linear range of detection extended over nearly three orders of magnitude (Fig. 2d). We also found that MacroGreen could recognize remaining fractions of auto-MARylated PARP10 that had been treated with side chain linkage specific ADP-ribosyl glycohydrolases (Supplementary Fig. 4). Together, these results suggest that MacroGreen recognized ADP-ribosylated targets independent of the side chain linkage of the modification.

Finally, we established that MacroGreen could bind to ADP-ribosylated histones, which are known targets for PARP10. Specifically, MacroGreen detected PARP10-MARylated histone proteins H2B, H3.1, H3.2, H3.3 and H4 (Fig. 2e). To conclude, our results show that MacroGreen has high affinity for PARP10 auto-MARylated at a number of side chains and can be used to quantify MAR- and PARylation of various PARP enzyme targets with ADP-ribose linkages at various amino acid side chains.

**MacroGreen can be used to evaluate small-molecule inhibition of PARP enzymes**. Having established that MacroGreen fluorescence increases in a linear manner over a wide range of concentrations of MAR- and PARylated target proteins, we evaluated enzyme inhibition assays based on MacroGreen detection. This was important, since a MacroGreen based screening assay may help facilitate drug discovery and development. Inhibition of PARP1 by Talazoparib and of PARP10 by PJ34, two widely used PARP inhibitors, was verified using Western blotting (Fig. 3a). MacroGreen overlay assays in 96-well plates reflected the pattern of inhibition (Fig. 3b) and were consistent with in vitro $IC_{50}$ values.[35] Amenability to high-throughput screening is an important requirement for ADP-ribosylation assays. Therefore, we also established standard protocols and assay descriptors for this method (Supplementary Note 1).

**MacroGreen marks ADP-ribosylated sites in fixed and permeabilized cells**. To analyze if MacroGreen could be used to detect ADP ribosylation in a physiological process, we turned to identification of DNA damage sites by fluorescence microscopy using MacroGreen as a fluorescent probe. The LacR-FokI system was used to induce DNA double-strand breaks within a single genomic locus in U2OS cells. This cell line stably expresses the mCherry-LacI-FokI nuclease fused to a destabilization domain and a modified estradiol receptor, allowing inducible nuclease expression after administration of the small molecules Shield1 ligand and 4-hydroxytamoxifen (4-OHT).[36]

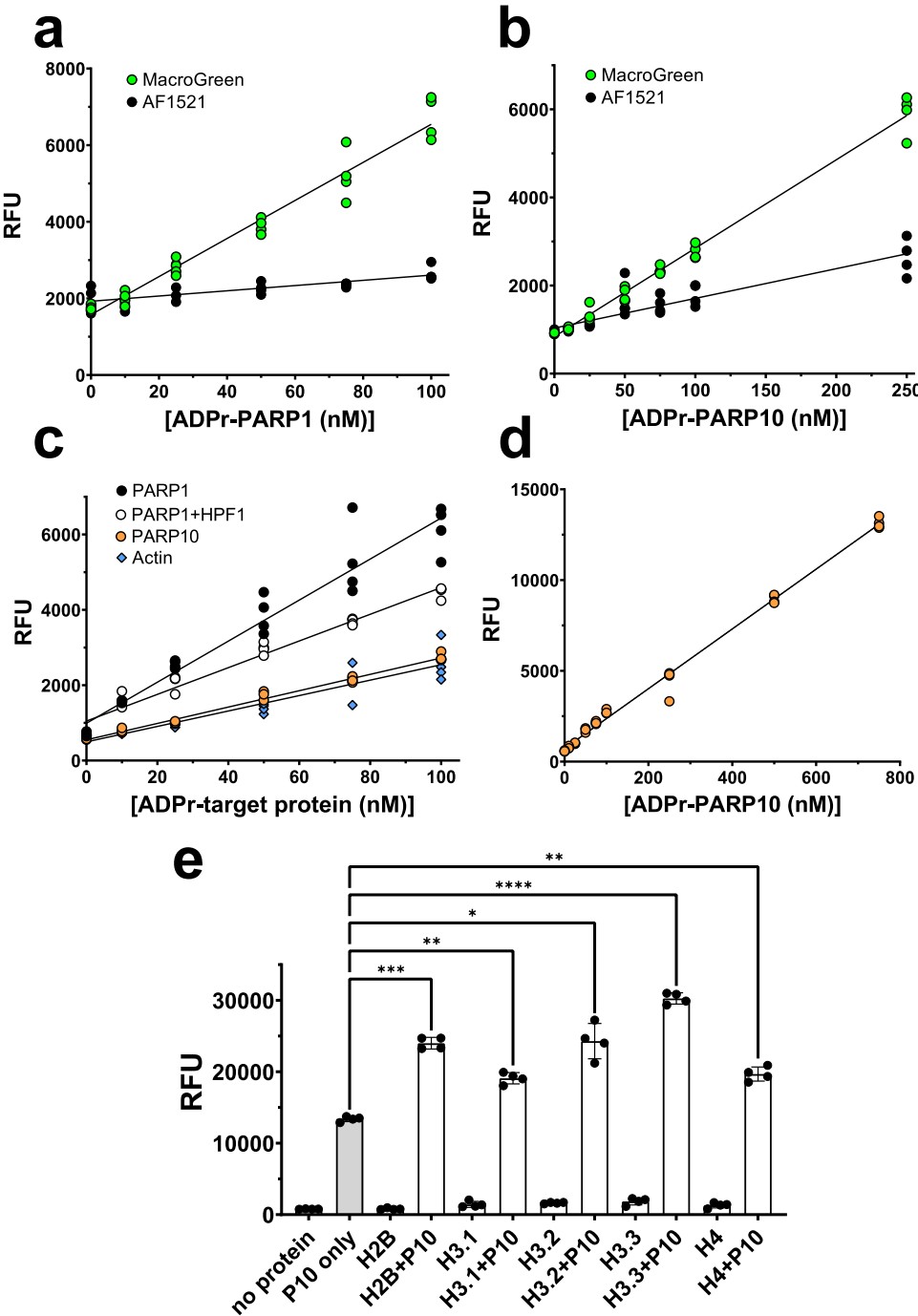

**Fig. 2 MacroGreen can detect an array of ADP-ribosylated substrates.** Comparison between MacroGreen and the wild type Af1521-GFP fusion binding to either PARylated PARP1 (**a**) or MARylated PARP10 (**b**) coated plates. In total, 50 µl reactions in 96-well Nunc MaxiSorp™ plates; detection with 1 µM MacroGreen or Af1521-GFP at room temperature **c** Examples of the linear assay range using MacroGreen detection of PARP1 PARylation predominantly at acidic side chains; PARP1 PARylation predominantly at serine side chains in the presence of HPF1; PARP10 MARylation at various side chains; or C2I MARylation of actin at Arg177. **d** Extended linear range of MAR detection on PARP10 under the same conditions. **e** Detection of MAR on five histone proteins (H2B, H3.1, H3.2, H3.3 and H4 as indicated) in the presence of PARP10. PARP10 MARylation alone (gray) serves as reference. Negative control reactions contained $NAD^+$ (1 mM) but not PARP10. Statistical significance of the data was calculated using a one-way ANOVA test; error bars indicate S.D. All panels – $n = 4$.

MacroGreen was able to bind to nuclear ADP-ribosylation induced by specific double-strand breaks produced by the FokI nuclease (Fig. 4a and Supplementary Fig. 5) and co-localized with yH2AX, a standard marker of DNA double-strand breaks.[37] After administration of PARP1 inhibitor, no MacroGreen signal was detected at the DNA damage site, consistent with MacroGreen recruitment being dependent on PARP1-dependent ADP-

ribosylation. The residual nuclear and cytosolic staining with MacroGreen in the presence of PARP inhibitor is consistent with detection of general ADP-ribosylation not specifically induced by DNA damage. To further confirm the specificity of the fluorescent signal produced, a GFP control was used. No staining of DNA lesions was observed in control experiments using GFP protein instead of MacroGreen

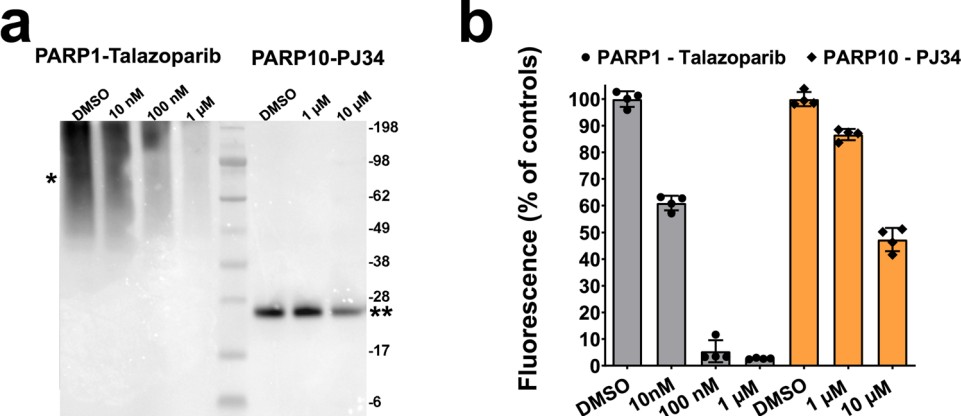

**Fig. 3 MacroGreen can be used to evaluate small-molecule inhibition of PARP enzymes. a** Verification of PARP1 inhibition by Talazoparib and of PARP10 inhibition by PJ-34 using Western blotting. **b** Quantification using MacroGreen of PARP1 and PARP10 inhibition. $n = 4$; error bars indicate S.D.

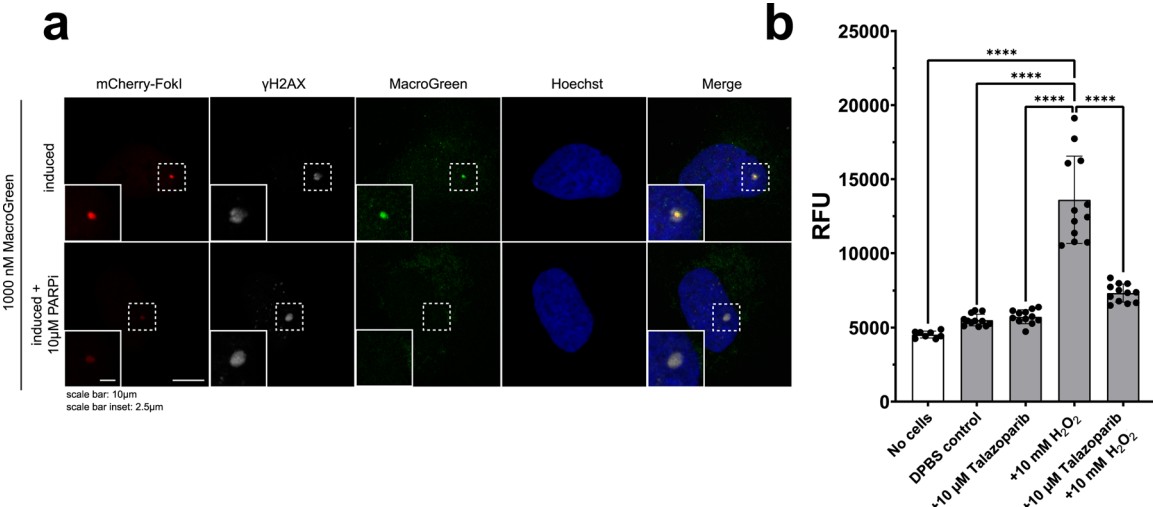

**Fig. 4 MacroGreen can detect ADP-ribosylated sites in fixed and permeabilized cultured cells. a** MacroGreen staining of DNA damage sites in U2OS DNA double-strand break reporter cells, verified by staining for γH2AX. Control experiments contained a PARP1 inhibitor. Scale bars: Main panel, 10 μm; inset, 2.5 μm. **b** MacroGreen fluorescence detected after staining of HEK293T cells grown in 96-well plates and treated to induce DNA damage as indicated. $n = 12$; error bars indicate S.D.

(Supplementary Fig. 6). These experiments show that Macro-Green binding occurred at the sites of DNA damage and was dependent on PARP1 activity.

To further explore the utility of MacroGreen as a tool for detection of PARP physiological activities, we documented MAR- and PARylation levels using a plate reader in fixed and permeabilized HEK293T cells treated to induce DNA damage in the absence or presence of Talazoparib, a potent inhibitor of PARP1. When cells were treated with 10 mM $H_2O_2$ for 10 min, the levels of ADP-ribosylation detected by MacroGreen fluorescence were consistently higher than in plate wells containing untreated cells (Fig. 4b). Control experiments showed that free GFP protein did not produce fluorescence above background levels (Supplementary Fig. 7). When cells were co-treated with 10 μM Talazoparib, the MacroGreen fluorescence levels were significantly lower than in cells treated to induce oxidative DNA damage (Fig. 4b). These results suggest that the increase in the ADP-ribosylation signal in response to $H_2O_2$ treatment was primarily due to PARP1 activity and can be readily detected by MacroGreen.

## Conclusions

Our mutagenesis approach identified a quadruple mutant of Af1521 with apparent high affinity ($K_{D,app} = 30$ nM) binding to ADP-ribosylated targets and a rate of ADP-ribosyl glycohydrolase activity reduced to half compared to the wild type domain ($k_{cat} = 0.075 \pm 0.007$ min$^{-1}$ vs. $0.153 \pm 0.018$ min$^{-1}$). With these properties, we detected stable fluorescence signals in several different assays. Overexpression in *E.coli* is exceptionally efficient and purification via immobilized metal ion affinity and size exclusion chromatography is straightforward, yielding around 100 mg of pure protein per liter of culture. Thus, it is our expectation that this simple and inexpensive research tool will facilitate studies into ADP-ribosylation where more elaborate methods are not an option.

## Methods

**Molecular cloning.** All Af1521 expression plasmids were made by subcloning synthetic cDNAs, codon optimized for bacterial expression (GeneArt; Thermo Fisher Scientific) into a pNIC28 derived vector[38] to obtain constructs encoding N-terminal hexahistidine and C-terminal eGFP fusions of the full length Af1521 protein variants. PARP1[662–1011], PARP10[1–1024] and PARP10[819–1007] constructs have been described before.[35] HPF1 expression plasmid pET30-HPF1-His-Sumo-Flag was contributed by Tom Muir[39] and obtained from Addgene (#111277). Human ARH3 expression plasmid (full length sequence in pET26)[40] was kindly contributed by Friedrich Koch-

Nolte (University Medical Center Hamburg-Eppendorf). The cDNAs encoding human MacroD2[21–245] and human TARG/C6orf130[1–152] were inserted into pNIC28 to obtain N-terminal hexahistidine fusions.[38]

**Protein purification**. All Af1521-GFP fusion constructs (eAf1521 and Af1521-c001 to -c008) were transformed and expressed in *Escherichia coli* BL21(DE3)T1R cells (SigmaAldrich) carrying the pRARE2 plasmid (Karolinska Institutet Protein Science Facility). A similar strain is available from Addgene (#26242). Cells were grown at 37 °C in 2-liter TunAir flasks in 750 ml Terrific Broth (TB) media supplemented with 50 µg ml$^{-1}$ kanamycin. At an $OD_{600}$ of 2.0, protein expression was induced with 0.5 mM IPTG at 18 °C for 16 h. Cells were harvested by centrifugation, resuspended in lysis buffer (50 mM HEPES, 500 mM NaCl, 10% glycerol, 0.5 mM TCEP, pH 8.0) supplemented with an EDTA-free protease inhibitor cocktail (SigmaAldrich) and DNAse (Sigma Aldrich), and lysed by a freeze-thaw cycle and sonication. The lysate was clarified by centrifugation and filtration (0.45 µm). For immobilized metal affinity chromatography (IMAC), clarified lysate was loaded onto a 5-ml HiTrap Talon Crude column (GE Healthcare) and after washing (lysis buffer, 10 column volumes), bound proteins were eluted with lysis buffer containing 300 mM imidazole. IMAC fractions contained highly concentrated protein which was further purified by size exclusion chromatography (SEC) using a Superdex-75 (16/60; GE Healthcare). Purified protein was concentrated by ultrafiltration and stored at −80 °C until use. MacroGreen production and purification were exceptionally efficient, with final yields of 100-200 mg of protein per liter of bacterial culture. Judged by SDS-PAGE and Coomassie staining, MacroGreen protein was >80% pure after IMAC, and >95% pure after SEC (Supplementary Fig. 8). PARP1 and PARP10 protein expression and purification were carried out as described.[35,41] All other hexahistidine tagged proteins were produced and purified as above, but using imidazole gradients in IMAC and either Superdex-75 or Superdex-200 in SEC. The *Clostridium botulinum* C2I toxin subunit, expressed as a glutathione-S transferase (GST) fusion construct and purified including removal of the GST tag,[34,42] was contributed by Prof. Holger Barth (University of Ulm, Germany). Bovine cytosolic β/γ-actin isoform mixture was purified from calf thymus.[43]

**ADP-ribosylation reactions for biotinyl-NAD assay**. PARP10 automodification was measured essentially as described before.[35,41] Briefly, 2% biotinylated NAD$^+$ (BPS Bioscience; final concentration 200 µM) was used in the ADP-ribosylation reactions, PARP10 was incubated in Nickel coated plates (Pierce), and protein-attached biotin-ADP-ribose was quantified with streptavidin-horseradish peroxidase and luminol reagent in a CLARIOstar multimode reader (BMG Labtech).

**ADP-ribosylation of samples for the plate-based MacroGreen assay**. For automodification of PARP1, 50 picomoles (1 µM in 50 µL) of PARP1 catalytic domain construct were incubated with 1 mM NAD$^+$ in reaction buffer (50 mM HEPES, 100 mM NaCl, 0.2 mM TCEP, 4 mM MgCl$_2$, pH 7.5) for 30 min at room temperature (RT; approximately 23 °C). For serial dilutions, the reaction was stopped by addition of Talazoparib (2 µM). The protein was diluted as indicated and dispensed into protein binding multi-well plates. For automodification of PARP10, 50 picomoles (1 µM in 50 µL) of the PARP1 catalytic domain or full-length protein were incubated with 1 mM NAD$^+$ in reaction buffer (as above) for 30 min at RT. For standard curves the reaction was serially diluted as indicated and dispensed into protein binding multi-well plates. For MARylation of histone proteins by PARP10, 10 picomoles (0.2 µM in 50 µL) of the

catalytic domain of PARP10 were co-incubated with 50 picomoles (1 µM in 50 µL) of histone protein (New England Biolabs cat. no. M2503, M2504, M2505, M2506, and M2507) in the presence of 1 mM NAD$^+$ in reaction buffer for 30 min at RT. For MARylation of actin by Clostridium toxin, 2.5 picomoles (0.05 µM in 50 µL) of the C2I toxin subunit were co-incubated with 50 picomoles (1 µM in 50 µL) of bovine non-muscle actin (β/γ-isoform mixture) in the presence of 1 mM NAD$^+$ in reaction buffer for 30 min at RT.

**Generic plate binding assay protocol**. In total, 50 µL of the MAR- or PARylated sample per well were dispensed into protein binding 96-well plate (MaxiSorp$^{TM}$ or PolySorp$^{TM}$, Nunc; Greiner Bio-One #655076; or similar) and incubated for 30 min at RT under constant shaking to allow protein binding to the plate. Unbound protein was removed with 3 washes of 15 s with 150 µL/well of TBST buffer (50 mM Tris-HCl, 150 mM NaCl, 0.1 % Tween-20, pH 7.5). Wells were blocked with 150 µL/well 1% (w/v) BSA solution in TBST buffer and incubation for 5 min at RT. Unbound BSA was removed with 3 washes of 15 s with 150 µL/well of TBST buffer. To quantify ADP-ribosylated protein, a suitable concentration of MacroGreen was determined for each target protein and plate by serial dilution between 40 to 5000 nM in TBST buffer. In total, 50 µL MacroGreen solution was added per well and incubated at RT under constant shaking for 5 min. Unbound MacroGreen was washed with 3 washes of 15 s with 150 µL/well of TBST buffer. GFP fluorescence was measured using a CLARIOstar multimode reader (BMG Labtech), using a 470-15 nm excitation filter and a 515-20 nm emission filter.

**Western blotting analysis of PARP1 and PARP10**. PARP1 and PARP10 catalytic domain automodification reactions were carried out in the presence of 10% biotinylated NAD$^+$ (BPS Bioscience; final concentration 100 µM) and different concentrations of the inhibitors PJ34 and Talazoparib. Proteins and SeeBlue® Plus2 Pre-Stained Protein Standard (Thermo Fisher Scientific LC5925) were resolved by SDS-PAGE electrophoresis and transferred to a PVDF membrane by wet transfer. The membrane was blocked with 1% BSA in TBST buffer before incubation with Streptavidin-HRP conjugate (Thermo Fisher Scientific) in TBST buffer. Super-Signal$^{TM}$ West Pico PLUS (Thermo Fisher Scientific) HRP substrate was used during the developing step. Images were taken using a PXi4 imaging system (Syngene).

**MacroGreen staining in DSB reporter cells**. U2OS DSB reporter cells stably expressing ER-mCherry-LacR-FokI-DD[36] (a gift from Roger Greenberg, University of Pennsylvania) were cultured in DMEM + GlutaMAX supplemented with 10% fetal bovine serum at 37 °C and 5% CO$_2$. FokI expression was induced for 5 h by 1 µM Shield1 (Clontech) and 1 µM 4-hydroxytamoxifen (4-OHT; Sigma-Aldrich). In control experiments, damage-induced PARylation was inhibited using PARP1 inhibitor KU0058948 (BPS Bioscience) at a final concentration of 10 µM. Cells were washed in 1 x phosphate-buffered saline (PBS) and fixed in 4 % PFA in PBS for 15 min at RT. Cells were permeabilized with 0.5% Triton X-100 and quenching was performed with 100 mM glycine in PBS for 10 min. Samples were blocked in wash buffer (WB; PBS containing 0.5% bovine serum albumin (BSA) and 0.05% Tween-20) for 30 min. Primary antibody (mouse anti-γH2AX; JBW301, Millipore) was diluted 1:1000 in WB and applied for 1 h at RT. Cells were washed once in WB and incubated with the indicated concentration of MacroGreen or GFP diluted in WB for 20 min at RT. Samples were washed three times in WB prior to addition of secondary antibody (anti-mouse AlexaFluor647; Life-Technologies). The cells were washed three times in WB and once in PBS and subsequently counterstained using 2 µg mL$^{-1}$ Hoechst (Hoechst 33342, Thermo Fisher Scientific) for 15 min.

Mounting on glass slides was done using homemade Mowiol/DABCO mounting medium. The samples were analyzed using a Zeiss LSM880 confocal microscope equipped with a 63x Plan-A (1.4 NA) oil-immersion objective and images acquired using Zen Black v2.1 software.

**Cell culture and DNA damage induction.** Human 293 T cells (a gift from Julian Walfridsson, Karolinska Institutet) were cultured in DMEM (Thermo Fisher Scientific) supplemented with 10% FBS (Thermo Fisher Scientific) and 1% penicillin/streptomycin (Thermo Fisher Scientific). Cells were grown at 37 °C and 5% $CO_2$. To induce DNA damage, 10 mM $H_2O_2$ in DPBS was added to cells for 10 min at 37 °C. To block the DNA damage-induced PARP1 activity, co-treatment with 10 μM Talazoparib (Sigma Aldrich) was used. As controls, cells only treated with either DPBS or 10 μM Talazoparib in DPBS were included in the experiment. After treatment, cells were washed in PBS and fixed in 4 % PFA in PBS for 15 min at RT. Cells were permeabilized with 0.5 % Triton X-100 followed by 100 mM glycine in PBS for 10 min. Wells were then washed 3 times with 150 μL/well of TBST buffer (50 mM Tris-HCl, 150 mM NaCl, 0.1 % Tween-20, pH 7.5) and blocked with 150 μL/well 1% (w/v) BSA solution in TBST buffer and incubation for 5 min at RT. Unbound BSA was removed with 3 brief washes of TBST buffer. 50 μL (1 μM) MacroGreen solution in TBST was added per well and incubated at RT under constant shaking for 5 min. A 1 μM solution of free GFP protein in TBST was used as control. Unbound MacroGreen or GFP were removed with 3 brief washes of 150 μL/well of TBST buffer. GFP fluorescence was measured using a CLARIOstar multimode reader (BMG Labtech), using a 470-15 nm excitation filter and a 515-20 nm emission filter.

**Statistics and reproducibility.** A one-way ANOVA with multiple comparison analysis by Dunnett´s test was performed to determine significance levels, using Prism version 9.0 (GraphPad). $P$ values are indicated by asterisks; $P \leq 0.05$ (*), $P \leq 0.01$ (**), $P \leq 0.001$ (***) and $P \leq 0.0001$ (****). Please see Supplementary Methods for further information.

**Availability of materials.** The MacroGreen expression plasmid is available through Addgene (ID 160665).

Reporting summary. Further information on research design is available in the Nature Research Reporting Summary linked to this article.

## Data availability

We declare that the data supporting the findings of this study are available within the paper and the associated Supplementary Information. The source data for the graphs and charts in the figures is given as Supplementary Data 1 and any remaining info can be obtained from the corresponding author upon reasonable request.

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

## Acknowledgements

We thank Tomas Nyman (Karolinska Institutet Protein Science Facility) for molecular cloning, Friedrich Koch-Nolte (University Medical Center Hamburg-Eppendorf) for the human ARH3 expression plasmid, Holger Barth (University of Ulm) for C2I toxin, Roger Greenberg (University of Pennsylvania) and Julian Walfridsson (Karolinska Institutet) for cell lines, Anna Rising (Karolinska Institutet) for making the Biacore equipment available, and Nina Kronqvist (Karolinska Institutet) for advice on Biacore experiments. This work was supported by the Swedish Research Council (2019-04871 to H.S. and 2016-02479 to N. P.D.), the Swedish Cancer Society (CAN 2017/492 to H.S. and 2018/693 to N.P.D.), the Swedish Childhood Cancer Fund (PR2018-0145 to H.S.) and Karolinska Institutet.

## Author contributions

H.S. and N.P.D. designed experiments, obtained funding, analyzed and interpreted the data; A.G.G-S. and L.K.H. designed and carried out experiments and interpreted the data; A.G.G.S. drafted the manuscript and composed the figures; L.K.H and N.P.D. participated in writing the manuscript and composing the figures; H.S. wrote the final manuscript version. All authors reviewed and approved submission.

## Funding

## Competing interests

The authors declare no competing interests.
