## [Peer Review File · Communications Biology]

Reviewers' comments:

Reviewer #1 (Remarks to the Author):

The manuscript by Gines Gaercia Saura et al. describes a new tool for detecting ADP-ribosylated proteins. They present improvements upon the established Af1521 macrodomain tool that is used for the detection and enrichment of ADP-ribosylated proteins. Af1521 is linked to a GFP tagged for easy detection and key mutations are made, both those present in eAf1521 (a newly described ADP-ribose binding protein with enhanced ADP-ribose affinity) and additional mutations that reduce ADP-hydrolase activity, which is a limitation of both Af1521 and eAf1521. This new tool is called MacroGreen. The authors then demonstrate the efficacy of MacroGreen in a plate setting (for inhibitor screening) and in a cell setting for detection of ADP-ribosylated proteins in fixed cells.

Overall, the manuscript is well written and the experiments are generally well controlled. The accessibility and ease of generation and use will likely encourage not only those in the field of ADP-ribosylation to use MacroGreen, but also those interested in entering this burgeoning field. A few minor suggestions/questions are detailed below:

1. The authors should consider moving 1 or 2 figures from the supplement to the main text (e.g. Figure 2 or Figure 5).
2. Supplementary Figure 2: The data used here is not sufficient to support that detection of ADP-ribose is independent of side chain linkage. The authors should remove this statement or perform experiments that directly address chain linkage specificity (e.g. PARP1 alone, which modifies on Glu/Asp, versus PARP1 + HPF1, which modifies on Ser).
3. Supplementary Figure 6: Can the authors provide the merged images? There is also staining by MacroGreen in sites besides the DNA damage site and in the presence of their PARP inhibitors. The authors should comment on this staining.
4. It would be ideal if the data in Table 1 and Table 2 are combined.
5. The authors state: "We conclude that MacroGreen overlay assays in microtiter plates are not useful for the quantification of ADP-ribosylation levels in cell lysates." Have the authors tried detection of ADP-ribose with MacroGreen using fixed cells instead of cell lysates? Would this circumvent the detergent problem and still be useful for screening PARP inhibitors?

Textual changes:

1. "Only recently, also an anti-MAR/PAR antibody has been presented." This sentence needs to be edited.
2. "An antibody that can specifically recognize mono-ADP-ribosylated targets is not available" – This statement should be amended to reflect recent work that shows antibodies detecting mono-ADP-ribosylation [Bonfiglio et al., Cell 183, 1086 (2020)].
3. "(Nunc MaxiSorp™ with medium binding plates (Greiner Bio-One #655076)" Parenthetical close is missing.

Reviewer #2 (Remarks to the Author):

COMMSBIO-20-3501-T by Saura et al. (Schüler)

"MacroGreen, a simple tool for detection of ADP-ribosylated proteins"

Summary

Saura et al. report that MacroGreen, an engineered version of Af1521 macrodomain fused to GFP, can be used to detect ADP-ribosylation by GFP fluorescence in a microplate reader, or in cells by

microscopy. They demonstrate that MacroGreen has reduced ADP-ribosyl glycohydrolase activity compared to eAf1521, a recently reported engineered version of Af1521, which has enhanced affinity for ADP-ribosylated proteins.

Strengths and Weaknesses:

Strengths: Overall, this study is interesting as it explores ways to enhance the affinity of the macrodomains for ADP-ribose, while reducing their hydrolase activity. The authors have performed various biochemical assays to measure the binding affinity of these macrodomains to ADP-ribose.

Weaknesses: Although the authors have developed a better version of the Af1521 macrodomain, this study lacks important controls, as noted below. Also, the authors did not compare the sensitivity of MacroGreen with wild-type AF1521 macrodomain in functional assays, such as ELISA and fluorescence microscopy. This limits our ability to understand if these tools perform better than the wild-type AF1521, or how much better.

Review

Major Comments

1. How does the binding of MacroGreen to MARYlated PARP-10 compare to its binding to MARYlated PARP-1 and PARYlated PARP-1?
2. In supplementary Figure 4, the authors demonstrate MacroGreen binding to PARP-10 substrates, such as histone proteins, by performing MARYlation reactions in microplates. Although they use histone proteins alone as controls, this does not exclude the possibility that the signal comes from automodification of PARP-10, not from the transmodification of histones. In this regard, it may be better to perform MARYlation reactions first and purify histone proteins, and then attach only histone proteins to the microplate for detection.
3. The authors showed the K_{cat} values in supplementary Table 2 and the curves in supplementary Figure 2b. But, is there a significant difference between Af1521-c007 and eAf1521 in these assays?
4. In supplementary Figure 4d, it may be better to add the basal condition without DNA damage as a control.
5. The authors measured the levels of DNA damage induced ADP-ribosylation using MacroGreen in supplementary Figure 5. Can the authors elaborate if they are able to distinguish MARYlation from PARYlation in these assays? It is not very clear since earlier they conclude that MacroGreen has high affinity for MARYlated proteins and PARP-10 substrates, but they observe a reduction of the fluorescence signal with a PARP-1 inhibitor. Do the authors see a reduction in fluorescence intensity of MacroGreen when the cells are treated with a pan MARYlation inhibitor?
6. In Figure 1c, it would be helpful if the authors presented Western blots to indicate the reduction of auto-modification of PARP-1 and PARP-10 by the inhibitors.
7. The authors should indicate how many times the experiments were repeated and the statistical significance for the bar graphs.

Minor Comments

1. The authors introduced mutations into AF1521 based on the outcomes observed in similar mutations of other hydrolases such as TARG and CHIKV nsP3. Are the structures of these macrodomains similar? It will be helpful to the readers if they include a schematic of how these regions compare between various macrodomains.
2. The authors should indicate the statistical significance in supplementary Figure 5? Give that the signal from H202+ Talazoparib is even lower than control group, it's better to add the Talazoparib treatment under basal condition as a control. Also, it's important to show the control of the same amount of protein used or same numbers of cells in this assay.
3. In supplementary Figure 7, the authors should to indicate if GFP and MacroGreen can penetrate the cells and nuclei equally. Does using higher levels of GFP or MacroGreen result in saturation of

fluorescence signals?

4. In the figure legend for Figure 1, panel c was marked as panel b.

Response to reviewers' comments:

Reviewer #1 (Remarks to the Author):

The manuscript by Gines Garcia Saura et al. describes a new tool for detecting ADP-ribosylated proteins. They present improvements upon the established Af1521 macrodomain tool that is used for the detection and enrichment of ADP-ribosylated proteins. Af1521 is linked to a GFP tagged for easy detection and key mutations are made, both those present in eAF1521 (a newly described ADP-ribose binding protein with enhanced ADP-ribose affinity) and additional mutations that reduce ADP-hydrolase activity, which is a limitation of both Af1521 and eAf521. This new tool is called MacroGreen. The authors then demonstrate the efficacy of MacroGreen in a plate setting (for inhibitor screening) and in a cell setting for detection of ADP-ribosylated proteins in fixed cells.

Overall, the manuscript is well written and the experiments are generally well controlled. The accessibility and ease of generation and use will likely encourage not only those in the field of ADP-ribosylation to use MacroGreen, but also those interested in entering this burgeoning field.

We thank this reviewer for their time, effort, and encouragement.

A few minor suggestions/questions are detailed below:

1. The authors should consider moving 1 or 2 figures from the supplement to the main text (e.g. Figure 2 or Figure 5).

As a response to this suggestion and the editorial request, we have made a major rearrangement of the manuscript including transferring figures from the SI to the main text.

2. Supplementary Figure 2: The data used here is not sufficient to support that detection of ADP-ribose is independent of side chain linkage. The authors should remove this statement or perform experiments that directly address chain linkage specificity (e.g. PARP1 alone, which modifies on Glu/Asp, versus PARP1 + HPF1, which modifies on Ser).

We agree that the data displayed in Supplementary Figure 4 (old Supplementary Figure 2) are only circumstantial evidence for side chain linkage independent detection of ADP-ribosylation. We have now performed the suggested experiment: We produced PARP1 ADP-ribosylated on serine residues (in the presence of HPF1) over a range of concentrations and quantified the modification using MacroGreen. The old and new data are displayed in Figure 2c. We also remind the reviewer of the detection of mono-ADP-ribosylated actin (Fig. 2c). With that, we show detection of ADP-ribosylation at Glu/Asp (PARP1 alone), Ser (PARP1 + HPF1), Arg (actin), and the mixture of all (PARP10). With these results and given the different chemical structures of these side chains, we believe it is correct to say: "Together, these results suggest that MacroGreen recognized ADP-ribosylated targets independent of the side chain linkage of the modification." (Line 147f).

3. Supplementary Figure 6: Can the authors provide the merged images?

We provide the merged images in Figure 4a of the main text and in Supplementary Figure 7.

There is also staining by MacroGreen in sites besides the DNA damage site and in the presence of their PARP inhibitors. The authors should comment on this staining.

We have addressed that staining by stating (line 181ff): “Residual nuclear and cytosolic staining with MacroGreen is consistent with detection of general ADP-ribosylation not specifically induced by DNA damage.”

4. It would be ideal if the data in Table 1 and Table 2 are combined.

We agree. The new Table 1 of the main text contains the protein construct designations, mutations they contain, and the glycohydrolase rates.

5. The authors state: “We conclude that MacroGreen overlay assays in microtiter plates are not useful for the quantification of ADP-ribosylation levels in cell lysates.” Have the authors tried detection of ADP-ribose with MacroGreen using fixed cells instead of cell lysates? Would this circumvent the detergent problem and still be useful for screening PARP inhibitors?

We thank the reviewer for this suggestion. Despite relatively high noise in the fluorescence measurements caused by autofluorescence of the plasticware, this method required fewer manipulation and produced reliable data, thereby expanding the range of methods that this detection reagent is useful for. The results of the suggested experiment are displayed in Figure 4b (with negative control experiments and a control of even numbers of cells per plate well in Supplementary Figure 9a). They confirm the observations made by fluorescence microscopy; significant MacroGreen staining is obtained after treating cells with DNA damage inducing H₂O₂ and lost by simultaneous addition of PARP inhibitor.

Textual changes:

1. “Only recently, also an anti-MAR/PAR antibody has been presented.” This sentence needs to be edited.

2. “An antibody that can specifically recognize mono-ADP-ribosylated targets is not available” – This statement should be amended to reflect recent work that shows antibodies detecting mono-ADP-ribosylation [Bonfiglio et al., Cell 183, 1086 (2020)].

These two sentences have been merged and the content corrected. The sentence now reads (line 42f): “Recently, an antibody that can recognize both MAR and PAR has been presented⁸ and antibodies that specifically recognize MARYlated targets have been produced.⁹”

3. “(Nunc MaxiSorp™ with medium binding plates (Greiner Bio-One #655076)”
Parenthetical close is missing.
This has been corrected.

Reviewer #2 (Remarks to the Author):

COMMSBIO-20-3501-T by Saura et al. (Schüler)
“MacroGreen, a simple tool for detection of ADP-ribosylated proteins”

Summary

Saura et al. report that MacroGreen, an engineered version of Af1521 macrodomain fused to GFP, can be used to detect ADP-ribosylation by GFP fluorescence in a microplate reader, or in cells by microscopy. They demonstrate that MacroGreen has reduced ADP-ribosyl glycohydrolase activity compared to eAf1521, a recently reported engineered version of Af1521, which has enhanced affinity for ADP-ribosylated proteins.

Strengths and Weaknesses:

Strengths: Overall, this study is interesting as it explores ways to enhance the affinity of the macrodomains for ADP-ribose, while reducing their hydrolase activity. The authors have performed various biochemical assays to measure the binding affinity of these macrodomains to ADP-ribose.

Weaknesses: Although the authors have developed a better version of the Af1521 macrodomain, this study lacks important controls, as noted below. Also, the authors did not compare the sensitivity of MacroGreen with wild-type AF1521 macrodomain in functional assays, such as ELISA and fluorescence microscopy. This limits our ability to understand if these tools perform better than the wild-type AF1521, or how much better.

We wish to thank also this reviewer for the appraisal of our work and the time spent.

Since not all points raised under “Weaknesses” are repeated below, we wish to draw the reviewer’s attention to our new Figure 2 panels a,b, which feature the side-by-side comparison between MacroGreen and the wild type Af1521-GFP protein.

Review

Major Comments

1. How does the binding of MacroGreen to MARYlated PARP-10 compare to its binding to MARYlated PARP-1 and PARYlated PARP-1?

A quantitative comparison of the detection of MAR- and PARYlation is presented in what is now Figure 2c. PARP1 is not a MARYlating enzyme by nature; therefore, we find it more meaningful to conduct this comparison using physiologically MARYlating enzymes and several of their substrates.

2. In supplementary Figure 4, the authors demonstrate MacroGreen binding to PARP-10 substrates, such as histone proteins, by performing MARYlation reactions in microplates. Although they use histone proteins alone as controls, this does not exclude the possibility that the signal comes from automodification of PARP-10, not from the transmodification of histones. In this regard, it may be better to perform MARYlation reactions first and purify histone proteins, and then attach only histone proteins to the microplate for detection.

The reviewer raises the possibility that any of 5 different histone proteins stimulate PARP10 auto-MARYlation activity without being a target themselves. This has no published prevalence; but side-by-side modification of both PARP10 and histones (on the same gels/membranes) does. We have attempted to execute the experiment as suggested by the reviewer; but, presumably due to the chemical nature of the histone proteins, it was not possible to purify PARP10 away from modified histones without a loss of protein that was difficult to control and detrimental to quantification. Instead, we have used an electrophoresis method to visualize simultaneous PARP10 auto- and histone MARYlation (PMID: 33804157). The result, displayed below, shows that PARP10 auto-MARYlation is roughly similar regardless of whether a histone protein is present or not. Histone-4 could not be resolved owing to its high charge-to-size ratio on these gels; but the gel shows that the PARP10 automodification in the histone-4 lane is similar as in the other PARP10 containing lanes.

3. The authors showed the Kcat values in supplementary Table 2 and the curves in supplementary Figure 2b. But, is there a significant difference between Af1521-c007 and eAf1521 in these assays?

We provide mean values and standard deviations as well as goodness of fit for each value and we provide the numbers of duplicates and technical replicates. This is according to standard operating procedure in the determination of this type of pseudo-first order enzymatic rate data. We are unaware of any suitable statistical model for testing such data to obtain a further measure of significance.

4. In supplementary Figure 4d, it may be better to add the basal condition without DNA damage as a control.

We agree with the referee and have performed this additional control in the new version of the experiment in question: Please see Figure 4b and Supplementary Figure 9a.

5. The authors measured the levels of DNA damage induced ADP-ribosylation using MacroGreen in supplementary Figure 5. Can the authors elaborate if they are able to distinguish MARYlation from PARYlation in these assays? It is not very clear since earlier they conclude that MacroGreen has high affinity for MARYlated proteins and PARP-10 substrates, but they observe a reduction of the fluorescence signal with a PARP-1 inhibitor. Do the authors see a reduction in fluorescence intensity of MacroGreen when the cells are treated with a pan MARYlation inhibitor?

This may be a misunderstanding: We conclude (and Figure 2 of the main text shows) that MacroGreen does not discriminate between MAR and PAR. Also, the main cellular response to DNA damage is PAR. We have added an explanatory comment (line 180ff) also in response to Reviewer 1. – A pan-MARYlation inhibitor is not available as far as we know.

6. In Figure 1c, it would be helpful if the authors presented Western blots to indicate the reduction of auto-modification of PARP-1 and PARP-10 by the inhibitors.

This is common knowledge in the PARP field, in particular concerning these two widely used compounds; but we agree fully with the reviewer that this will be a meaningful addition to a paper directed at a general audience and have added this experimental result as panel a of the new Figure 3 of the main text.

7. The authors should indicate how many times the experiments were repeated and the statistical significance for the bar graphs.

Statistical information has now been added to all graphs.

Minor Comments

1. The authors introduced mutations into AF1521 based on the outcomes observed in similar mutations of other hydrolases such as TARG and CHIKV nsP3. Are the structures of these macrodomains similar? It will be helpful to the readers if they include a schematic of how these regions compare between various macrodomains.

We have now added Supplementary Figure 1 explaining the structural basis of the design of site-specific mutations.

2. The authors should indicate the statistical significance in supplementary Figure 5? Give that the signal from H202+ Talazoparib is even lower than control group, it's better to add the Talazoparib treatment under basal condition as a control. Also, it's important to show the control of the same amount of protein used or same numbers of cells in this assay.

This experiment is now replaced; please see also our reply to comment 4 above. Statistical significance is indicated in the new experimental data (Figure 4b and Supplementary Figure 9a).

3. In supplementary Figure 7, the authors should to indicate if GFP and MacroGreen can penetrate the cells and nuclei equally.

Penetration into semipermeabilized, fixed cells is generally thought to be a function of molecular size. Hence, there is no reason to believe that the lack of staining with plain GFP, which functions as a negative control, is due to reduced penetration compared to MacroGreen.

Does using higher levels of GFP or MacroGreen result in saturation of fluorescence signals?

No; but background staining increased. This is consistent with our biochemical analysis which did not warrant a higher working concentration for staining of physiological target sites.

4. In the figure legend for Figure 1, panel c was marked as panel b.

This figure has been re-made and corrected.

REVIEWERS' COMMENTS:

Reviewer #1 (Remarks to the Author):

The authors have sufficiently addressed my concerns

Reviewer #2 (Remarks to the Author):

The authors have done a good job of addressing my previous concerns. I don't have any additional comments.